# Therapeutic Targeting of RNA Splicing in Cancer

**DOI:** 10.3390/genes14071378

**Published:** 2023-06-29

**Authors:** Elizabeth A. Bonner, Stanley C. Lee

**Affiliations:** 1Molecular and Cellular Biology Graduate Program, University of Washington, Seattle, WA 98195, USA; eab83@uw.edu; 2Translational Science and Therapeutics Division, Fred Hutchinson Cancer Center, Seattle, WA 98109, USA; 3Department of Laboratory Medicine and Pathology, University of Washington School of Medicine, Seattle, WA 98195, USA

**Keywords:** alternative splicing, splicing factor mutations, spliceosome inhibitors, anti-sense oligonucleotides, immuno-oncology therapies

## Abstract

RNA splicing is a key regulatory step in the proper control of gene expression. It is a highly dynamic process orchestrated by the spliceosome, a macro-molecular machinery that consists of protein and RNA components. The dysregulation of RNA splicing has been observed in many human pathologies ranging from neurodegenerative diseases to cancer. The recent identification of recurrent mutations in the core components of the spliceosome in hematologic malignancies has advanced our knowledge of how splicing alterations contribute to disease pathogenesis. This review article will discuss our current understanding of how aberrant RNA splicing regulation drives tumor initiation and progression. We will also review current therapeutic modalities and highlight emerging technologies designed to target RNA splicing for cancer treatment.

## 1. Introduction

RNA splicing is a fundamental mechanism of gene regulation in eukaryotes whereby premature mRNA (pre-mRNA) molecules are processed to form mature mRNA transcripts for protein translation. Most multi-exon genes undergo alternative splicing (AS), which generates multiple mature mRNA molecules to diversify the proteome and contributes to fundamental cellular processes, including cellular differentiation, development, and cell death. It is now widely appreciated that AS dysregulation is a key hallmark of multiple human diseases, including neurodegeneration, immune disorders, and cancer. Here, we review the basic mechanisms of splicing and how neoplastic cells co-opt splicing machinery in tumor initiation and progression. We will also review therapeutic targeting of splicing dysregulation with small molecules and emerging technologies.

## 2. Regulation of Splicing

RNA splicing is a complex and highly regulated process involving the removal of introns and the ligation of exons to produce mature mRNAs for protein translation. This is mediated by the spliceosome, a large complex consisting of ribonucleoproteins (RNPs) and small nuclear RNAs (snRNAs). Landmark studies in the last few years have increased our understanding of the structure and function of the eukaryotic spliceosome and are reviewed in [1]. The major spliceosome, which consists of five small nuclear RNPs (snRNPs), U1, U2, U4, U5, and U6, is responsible for removing ~99% of human introns, while the U5, U11, U12, U4atac, and U6atac snRNPs are responsible for minor intron splicing. Splicing catalysis is initiated when the spliceosome complex recognizes *cis*-regulatory sequences in the pre-mRNA such as the GU- and AG-dinucleotide sequences on the 5’ and 3′ splice sites (ss), respectively, the polypyrimidine tract, and the branchpoint sequence (BPS) (Figure 1A). The U1 snRNP binds the 5′ss, followed by the binding of splicing factor 1 (SF1) to the BPS located proximal to the 3’ss. The U2 auxiliary factors (U2AF1/U2AF2 heterodimer) recognize the 3’ss and the polypyrimidine tract. The distinction of the 3′ splice site is reinforced by the polypyrimidine tract, which serves as an essential signal for recruiting additional *trans*-acting factors to the 3’ splice site. Following the establishment of the early complex (complex E), the U2 snRNP, which contains the splicing factor 3b subunit 1 (SF3B1), displaces SF1 at the branchpoint via base pairing with U2 snRNAs and interacts with the U2AFs to form complex A. This is followed by the recruitment of the U4/U5/U6 tri-snRNP to form the activated B and catalytically active spliceosome (B*) complex, which executes the first trans-esterification reaction, followed by the second trans-esterification reaction by the catalytically active C (C*) complex. At the final step of splicing catalysis, the spliceosome components and the intron lariat dissociate from the ligated exons, forming a mature mRNA molecule (Figure 1B). Splicing can be further regulated by *trans*-acting RNA binding proteins (RBPs), including the serine/arginine (SR) and the heterogenous nuclear ribonuclear protein (hnRNP) family proteins, which possess the ability to promote or repress splicing by the sequence-specific recognition of *cis*-elements known as the exonic splicing enhancers (ESEs), intronic splicing enhancers (ISEs), exonic splicing silencers (ESSs), and intronic splicing silencers (ISSs) (Figure 1A), which are reviewed extensively here [2]. Together, the combination of *cis*-elements and *trans*-acting factors dictate the final usage of splice sites, resulting in a single gene that encodes multiple distinct protein isoforms.

## 3. Altered Splicing Factor Expression in Cancer

In the last decade, multiple studies revealed that the splicing machinery could be corrupted by cancer cells for disease initiation and progression. This includes altered expression and somatic mutations in *trans*-acting core splicing factors and synonymous mutations in *cis*-regulatory elements that inhibit the productive splicing of tumor suppressor genes [3,4]. Similarly, altered expression levels of multiple splicing factors have been observed in various solid tumors, resulting in the widespread dysregulation of alternative splicing patterns (summarized in Table 1). One of the most well-studied pro-tumorigenic splicing factors is SRSF1 [5,6], which is a part of the serine/arginine-rich (SR) protein family. SRSF1 is overexpressed in many cancers, including breast, lung, and colon, via copy number gain and altered expression regulation. Overexpression of SRSF1 promotes the growth of breast cancer cells and drives the alternative splicing of isoforms associated with various cancer hallmarks such as apoptosis (e.g., BIN1 and BIM), proliferation (e.g., RON, MKNK2, and S6K1), and DNA damage response (e.g., PTPMT1) [5,7,8]. SRSF1 is a known transcriptional target of MYC; SRSF1 overexpression collaborates with MYC in tumorigenesis in vitro and in vivo, in part, via an increased activation of the mTORC1 signaling pathway and protein translation. Other commonly mis-expressed splicing factors in solid tumors include multiple SR proteins, hnRNP proteins, and members of other RNA binding proteins, including RBM5, RBM10, and RBFOX2 (reviewed extensively here [9]).

## 4. Recurrent Mutations in Splicing Factors in Cancer

A decade ago, several landmark studies identified recurrent mutations in core splicing factors in myelodysplastic syndromes [10] (MDS) and other additional malignancies, including chronic lymphocytic leukemia (CLL), acute myeloid leukemia (AML), myeloproliferative neoplasms (MPN), uveal melanoma, pancreatic ductal adenocarcinoma, lung adenocarcinoma, and breast cancers (reviewed in [11]). Genes most frequently targeted for mutations are found in *SF3B1*, *SRSF2*, *U2AF1*, and *ZRSR2*. Mutations in *SF3B1*, *SRSF2*, and *U2AF1* occur exclusively as heterozygous missense mutations at “hot-spot” regions (Figure 2), whereas *ZRSR2* mutations are scattered across the gene and are predicted to confer a loss-of-function. Recently, mutations targeting U1, U2, and U11 snRNAs have been found in CLL and medulloblastoma (reviewed in [12]). Taken together, these studies provide strong evidence linking splicing perturbations to cancer pathogenesis. Even though the spliceosome machinery contains more than 180 proteins, the reason as to why there are only a few frequent targets of somatic mutations in cancers remains an open question. The following section will review how mutations in splicing factors affect normal splicing and the potential functional role of these mutations in myeloid neoplasms. We will then discuss the potential for targeting these mutations or reversing their effects with splicing modulators in cancer.

### 4.1. SF3B1 Mutations

SF3B1 is a component of the U2 snRNP that binds to the branchpoint in the early stages of pre-spliceosome formation, and is involved in recognizing the majority of 3’ss [13]. Transcriptomic analyses revealed that cancer-associated *SF3B1* mutations are generally associated with the usage of alternative 3’ (a3′ss) ~10–30 nucleotides upstream of the canonical 3’ss. The region around the cryptic 3’ splice site coincides with the enrichment of adenosines that also appear to have a stronger base-pairing affinity with the cognate U2 snRNA relative to the region around the canonical BPS. Structural analyses suggest that mutant SF3B1 may alter the charge and shape of the corresponding amino acid residues, which disrupts the interaction with pre-mRNA by approximately 10–30 nucleotides, consistent with bioinformatic predictions [14,15,16,17].

Mutations in *SF3B1* are enriched in a specific MDS subtype known as refractory anemia with ring sideroblasts (RARS), characterized by dysplastic erythroblasts with abnormal iron accumulation in the mitochondria that manifests as a “ring” of blue granules. Most of the mutations in *SF3B1* are clustered near the HEAT repeat domains 4 to 7 (HR4–HR7), with the most frequently mutated residues being K700 and K666 in MDS and CLL; while mutations in the R625 position are the most commonly occurring allele in uveal melanoma. The functional relevance of these distinct mutations to disease subtypes remains unclear and is an interesting area of focus for future studies.

Multiple studies predicted that mutant SF3B1-induced a3’ss usage results in the introduction of premature termination codons (PTCs), resulting in the nonsense-mediated decay (NMD) of the target transcript. To date, thousands of aberrantly mis-spliced transcripts have been identified across multiple cancer types, illustrating the robust effect of SF3B1 mutations on splicing. However, only a few targets have been causally implicated in disease phenotypes, including ABCB7, TMEM14c, ALAS2 in heme biosynthesis [18], BRD9 in the initiation and maintenance of solid tumors [19], and PPP2R5A in MYC regulation [20]. Overall, while global transcriptomic studies are powerful tools for inferring direct targets of aberrant splicing, these studies also highlight current challenges associated with identifying the functionally relevant and causative mis-splicing events that drive specific disease phenotypes.

### 4.2. U2AF1

*U2AF1* is mutated in ~15% of MDS, ~10% of CMML, and ~10% of secondary AML (s-AML) patients and is associated with poor prognosis. It is also found in a subset of pancreatic ductal adenocarcinomas and non-small cell lung adenocarcinomas. U2AF1/2 heterodimer recognizes the AG-dinucleotide at the 3’ss during the early steps of splicing catalysis in a sequence-specific manner [21]. *U2AF1* mutations are found in two hotspots, S34 and Q157, located within the zinc finger domains. Mutant U2AF1 affects splicing at the 3’ss: the S34 allele is associated with increased cassette exon inclusion if the nucleotide preceding the 3′ss is C/A over T, while the Q157 allele preferentially excludes exons containing A and includes exons containing G in the +1 position of the 3′ss [22]. Several mis-spliced targets include H2AFY, BCOR, ATR, and GNAS; aberrant H2AFY and STRAP isoforms are associated with myeloid-biased hematopoietic differentiation in CD34+ hematopoietic progenitors in ex vivo colony assays [23]. Further functional validation is needed to identify additional disease-causing isoforms.

### 4.3. SRSF2

*SRSF2* mutations are found in ~50% of chronic myelomonocytic leukemia (CMML), ~20% of MDS, and ~15% of AML patients and ~3–5% of healthy individuals with clonal hematopoiesis (CH). The presence of *SRSF2* mutation in MDS is often associated with poor prognosis and a higher risk of transformation to acute leukemia [12]. SRSF2 belongs to the serine/arginine-rich (SR) protein family and is involved in exon inclusion by binding RNA via the RNA recognition motif (RRM). Substitution at the proline 95 region of SRSF2 alters its RNA binding affinity in a sequence-specific manner such that wildtype SRSF2 binds C-rich and G-rich motifs in the ESE with similar affinity, while mutant SRSF2 prefers C-rich motifs and suppresses G-rich motifs [24]. Mutant SRSF2 mis-splices several key targets, including chromatin modifiers EZH2, BCOR, transcriptional regulator INTS3, and cell death regulator CASP8 [25].

### 4.4. ZRSR2

Somatic *ZRSR2* mutations are scattered across the coding region, typically occurring as frameshift indels, splice site, or nonsense mutations, and are predicted to disrupt the open reading frame (Figure 2). As an X-linked gene, the mutations of *ZRSR2* are found predominantly in male patients in ~10% of MDS and ~5% of CMML cases [12]. As a core component of the minor spliceosome, ZRSR2 is responsible for the splicing of minor introns, which accounts for ~1% of human introns. Interestingly, *ZRSR2* mutations can sometimes co-occur in patients with existing *SF3B1*, *SRSF2*, or *U2AF1* mutations. A loss of ZRSR2 is associated with an increased retention of U12-type containing introns, while the splicing of U2-type-containing introns were largely unaffected [26]. Functional work has identified that *ZRSR2* mutation is sufficient to promote the clonal advantage of bone marrow progenitors in vivo, partly via the mis-splicing of *LZTR1* [27].

### 4.5. Other Spliceosome Components

In addition to *SF3B1*, *U2AF1*, *SRSF2*, and *ZRSR2*, mutations in several spliceosome components have also been reported in both solid and hematologic malignancies, albeit at much lower frequencies. This includes genes encoding *DDX41* [28], *LUC7L2* [29], *PRPF8* [30], *PRPF40B* [10], *RBM10* [31], *U2AF2* [10], *SF3A1* [10], *SF1* [10], and *SRRM2* [32]. Moreover, hotspot mutations in U1, U2, and U11 snRNAs were recently observed in a subset of medulloblastoma, CLL, non-Hodgkin B-cell lymphoma (NHL), hepatocellular carcinoma, and pancreatic cancer patients [12]. Further functional validation experiments are required to dissect the relevance of these mutations to tumorigenesis.

### 4.6. Splicing in Metastasis and Treatment Resistance

Dissemination of tumor cells to colonize distal parts of the body is a hallmark of cancer and an indicator of poor prognosis and overall survival. Metastasis is driven, in part, by epithelial-to-mesenchymal transition (EMT) which enhances mobility, invasion and resistance to apoptotic stimuli. Research has found that dysregulation of RNA splicing is critical to disrupt the cell state to promote metastasis and extend cell survival. Epithelial splicing regulatory proteins (ESRPs), including ESRP1 and ESRP2, and RBFOX2 as have been identified as important regulators of metastasis [33]. ESRPs regulate RNA variants critical for epithelial identity and function [34]. Oncogenic dysregulation of ESRPs is facilitated by genetic and epigenetic alterations and post-transcriptional modifications, which modulate ESRP protein levels and splicing activity to govern the metastatic behavior of a variety tumor types (reviewed in [35]). Conversely, RBFOX2 governs mesenchymal splicing patterns [36]. Recent work has identified RBFOX2 as a metastatic suppressor in pancreatic cancer and correlated metastatic progression based on RBFOX2 expression levels and alternative splicing signatures in patient samples. Reduced RBFOX2 levels increased focal adhesion formation in vitro and increased metastatic lesion in vivo [37]. In addition to ESRPs and RBFOX2, other splicing factors are shown to govern alternative splicing events, which promote metastasis in a variety of solid tumors, including SF2/ASF (e.g., *RON* [38]), RBM4 (e.g., *MAP4K4* [39,40]), SRSF3 (e.g., *MAP4K4* [39], *HER2* [41]), and hnRNPs (e.g., *HER2*, *CD44*, *integrin β1* [40,41,42,43]).

Alternative splicing as a mechanism for drug resistance has also been widely reported. Changes in splicing can result from mutations in intragenic regions, which disrupt canonical splicing (e.g., *BIM*, *SLC29A1*, *dCK* [44,45]). Cancer cells can also alter the expression of splicing factors and splicing patterns to confer resistance, several of which are outlined in Table 2. Recent work has highlight alternative splicing as a mechanism of resistance to chimeric antigen receptor expressing T cell therapy (CART). While CART has been highly successful in treating several hematologic malignancies, a subset of patients experience relapse. In pediatric B-ALL, CD19-directed CART (CART-19) resistance has been shown to be mediated, in part, by changes in splicing. The skipping of exon 2 results in a reduced CD19 cell surface expression and abolishes CART-19 antigen recognition due to the removal of the FMC63 epitope [46]. Additionally, other variants, such as cassette exon skipping of exons 2, 5, or 6 result in NMD-mediated CD19 degradation or loss of cell-surface localization, respectively [47]. While some of these isoforms are predicted to occur at low levels prior to CART-19 therapy [48], research suggests that they may become the dominant splice variant due to alterations in splicing factor expression levels [46,47].

## 5. Therapeutic Targeting of RNA Splicing in Cancer

Given the importance of alternative splicing dysregulation in cancer initiation and progression, there has been significant interest in developing therapeutic strategies to target aberrant splicing in cancer. Various therapeutic modalities have been proposed and are at different stages of pre-clinical and clinical development ranging from small molecules (summarized in Figure 3) to oligonucleotide-based approaches (summarized in Figure 4). The following section summarizes current strategies used to target RNA splicing and explore novel technologies that are under pre-clinical development.

### 5.1. Targeting the Core Spliceosome with Small Molecule Inhibitors

FR901463, FR901464, and FR901465, isolated from *Pseudomonas* sp., were the first natural products shown to target core spliceosome components. Initial studies showed that these compounds had an antiproliferative effect in both murine and human solid tumor models. Based on the low IC50 value required to reduce tumor volume, FR901464 showed promise as a novel cancer therapeutic. Treatment with FR901464 led to enhanced SV40 promoter driven transcription, stalled cells in G1 and G2/M phase of the cell cycle, and induced inter-nucleosomal breakdown of chromatin [58]. Further analysis in vitro and in vivo showed that FR901464 treatment led to the production of a C-terminally truncated p27 isoform, resulting from aberrant splicing. To identify FR901464’s binding partners, a more chemically stable methyl ketal derivative, spliceostatin A (SSA), was developed. Through a series of biotin pull down assays, it was determined that both spliceostatin A and FR901464 associate with SF3B1 (Sap155), SF3B2 (SAP145), Sap130, and SF3B4 (SAP49), indicating that both molecules likely interact with the U2 snRNP, leading to impaired splicing [59]. Further mechanistic analysis of SSA revealed that, upon binding, SSA alters U2 snRNA branchpoint sequence preference by disrupting the SF3B1-RNA interaction. This hypothesis was supported by evidence from splicing microarrays showing that splicing patterns induced by SSA are partially recapitulated by the knockdown of SF3B1 [60]. The impact of SSA on splicing-independent functions has recently been demonstrated. This includes the premature cleavage, polyadenylation and cytoplasmic localization of a subset of transcripts, including the non-coding RNA MALAT1 [61].

Other natural products have been discovered, which inhibit the SF3B complex. GEX1 compounds, isolated from *Streptomyces* sp., were shown to induce apoptosis in solid tumors. GEX1A, one of several compounds tested, upregulated SV40 and cell cycle promoter driven transcription, stalled cell cycle progression at G1 and G2/M, and reduced CDK1 mRNA length in a dose-dependent manner [62]. In vitro, GEX1A treatment resulted in variable responses across multiple leukemic cell lines, which corresponded to overall survival in pre-clinical models [63]. Hasegawa et al. showed that GEX1A bound SF3B1, leading to splicing inhibition. As a consequence of GEX1A treatment, p27 was mis-spliced, which was hypothesized to contribute to cell cycle arrest [64]. In an attempt to understand GEX1A treatment variability, Hasegawa et al. found that the alternative splicing of pro-survival mRNAs encoding MCL-1 or pro-apoptotic mRNA encoding BIM were not predicters of resistance. It was determined, however, that resistant cell lines and pre-clinical models could be sensitized to GEX1A through treatment with BCL-xL inhibitors, which resulted in a synergistic increase in apoptosis [63].

Pladienolides are another class of natural products identified for their anticancer effects. Isolated from *Streptomyces* plantensis Mer-11107 [62], pladienolides (B–D), were first tested in a large cohort of drug-resistant solid tumor cell lines, and were shown to inhibit cell proliferation, disrupt cell cycle progression, and induced increased apoptosis. More excitingly, in vivo patient-derived xenograft (PDX) models treated with pladienolide B were shown to achieve complete remission following treatment [65]. Mechanistic evaluation revealed that pladienolides co-precipitated with U2 snRNP components SF3B1, TMG, SM BB’, D1, and U2B’. Additionally, pladienolides were also shown to associate with cyclin E, which form a complex with the U2 snRNA along with CDK2. Upon further evaluation, it was determined that pladienolide binds to SAP130 of the U2 snRNP to inhibit splicing [65,66].

While these initial studies were promising, the use of natural compounds remained limited. SF3B-targeting natural products are chemically complex and unstable in biological fluids, making them difficult to synthesize and limiting their clinical application. Efforts to improve natural products led to the development of several synthetic analogues, which could be used in research and further developed for clinical applications [64,67,68,69,70]. E7107, a synthetic analogue of pladienolide, was tested in 28 human tumor xenograft models. E7107-treated mice showed significant tumor regression, with a few animals achieving complete remission, some at a fraction of maximum tolerated dosage [64]. E7107 was the first splicing inhibitor to enter phase I clinical trials. In an open-label, single arm, dose escalation study, E7107 was administered to 26 patients with metastatic or locally advanced solid tumors experiencing relapse following treatment and for whom no other therapies were available. Unfortunately, this study was forced to terminate early due to two patients developing bilateral scotomas, leading to temporary or permanent visual loss [68,69].

More recently, an orally bioavailable small molecule splicing inhibitor, H3B-8800, was developed, which inhibited ATP-dependent 17S U2 snRNP complex formation, interfering with the association of SF3B complex with the BP sequence. Pre-clinical studies showed that H3B-8800 led to the preferential killing of splicing factor mutant cells in vitro and a lowered leukemic burden in *SF3B1* and *SRSF2* mutant PDX models [71]. These results are consistent with prior studies demonstrating that splicing factor mutations confer preferential sensitivity to spliceosome inhibitors [72,73,74]. In a phase I dose escalation clinical trial, 84 patients from the Unites States and Europe with hematologic malignancies (MDS, CMML, AML) underwent treatment with H3B-8800. While no complete or partial clinical response was observed (2006 IWG criteria), 15% of participants achieved some level of transfusion independence [75].

Recently, several small molecules targeting the U2AF heterodimer have emerged as novel modulators of RNA splicing. In a screen of 1593 compounds from the National Cancer Institute (NCI) Diversity Set V, NC 194308 was identified as a potent splicing inhibitor. In vitro studies showed that NC 194308 treatment increased the affinity of the SF1-U2AF1-U2AF2 complex to RNA but led to an accumulation of spliceosome A complex. It was determined that NC 194308 binds between U2AF2 RRMs, stabilizing it in a conformation favorable to RNA binding but prevents spliceosome assembly at the U2AF-dependant checkpoint for the polypyrimidine track. NC 194308 treatment perturbed alternative splicing of several well-characterized transcripts in vitro, events which were further perturbed in the presence of U2AF2 mutation [76]. Additionally, a phenothiazine derivative, 7,8-dihydroxyperphenazine, was recently identified as a U2AF inhibitor. In vitro splicing assays showed that 7,8-dihydroxyperphenazine inhibits spliceosome complex A-to-B transition by binding to the U2AF homology motif (UHM) on U2AF2, preventing interactions with the U2AF ligand motifs (ULMs) on SF1 and SF3B1 [76,77]. While further in vitro and pre-clinical testing is needed, preliminary data indicate that small molecules targeting the U2AF heterodimer may prove beneficial in the future.

Currently, the direct targeting of the spliceosome is not a viable method for the treatment of cancer. However, the development of drugs targeting the U2 snRNP have proven to be invaluable for scientific research. Compounds such as SSA are still routinely used to deconvolute the role-splicing plays in cancer development and maintenance. Thus, while the clinical application of splicing inhibitors has thus far proven to be fruitless, they may allow for the development of novel splicing-based treatment modalities for cancer.

### 5.2. Targeting Splicing Regulatory Proteins

Given the disappointing results from clinical trials aiming to directly target the spliceosome, leverage splicing regulatory proteins as an orthogonal therapeutic approach may prove to be advantageous. Splicing factor phosphorylation regulates their biochemical activity and sub-cellular localization. The altered expression of splicing regulatory proteins is observed across cancer types and are potential targets for anticancer therapies. Serine-rich protein kinases (SRPKs) and CDC-like kinases (CLKs), as well as the dual specificity tyrosine-regulated kinases (DYRKs) are well-known kinases that regulate the localization of splicing factors and alternative splicing.

SRPKs constitute an evolutionarily conserved subfamily of serine-threonine kinases that phosphorylate serine residues in serine-arginine/arginine-serine dipeptide motifs. SRPKs are found dispersed throughout the cell, and both expression and nuclear/cytoplasmic localization are tightly regulated by homeostatic pathways (reviewed in [78]). Mechanistically, both cytoplasmic and nuclear fractions assist in splicing modulation—cytoplasmic SRPKs phosphorylate newly synthesized SR proteins to facilitate nuclear translocation, and nuclear SRPKs are almost exclusively associated with ATP-dependent SR protein phosphorylation. Work published by Siqueira et al. demonstrated that SRPK protein expression varies across leukemia cell lines, with the most marked increase occurring in the cell lines of lymphoid origin [79]. In vitro, treatment with the SRPK1/2 dual inhibitor, SRPIN340, reduced cell viability regardless of SRPK expression levels and modulated MAPK and AKT signaling by changing the expression or splicing patterns of MAP2K1 and MAP2K2 or VEGF and FAS, respectively. Additionally, SRPK1 has been identified as a potential therapeutic vulnerability in a CRISPR dropout screen in AML [80].

The CLK family, which comprises CLK1-4, collaborates with SRPKs to adjust the degree of phosphorylation of RS dipeptides on SR proteins to modulate alternative splicing. Changes in CLK expression and activity are associated with cancer development and progression, and both depletion and chemical inhibition have been shown to alter splicing and decrease cell proliferation. The orally bioavailable pan-CLK inhibitor, T-025, exhibited anti-tumor activity in solid tumor xenografts and modulated the phosphorylation of SR proteins. T-025 treatment was shown to increase exon skipping across a variety of solid tumor lines in a dose-dependent manner and exhibited significant anti-tumor efficacy [81]. CTX-712, another CLK inhibitor, was recently developed and tested in the context of SRSF2 mutant hematopoietic malignancy. SRSF2 mutant MDS and AML PDX models showed a significant response to CTX-712, with many mice achieving complete remission [82], prompting the initiation of a currently ongoing multicenter, single-arm dose, phase I clinical trial for patients with hematologic malignancy (NCT05732103) [83]. To date, 18 unique CLK inhibitors have been investigated for their anti-tumor activity (reviewed in [84]).

Of the five members of the DYRK family (DYRK1A, 1B, -2, -3, -4), only DYRK1A is known to localize to nuclear speckles where it phosphorylates SR proteins and SF3B1 [85]. An increased gene dosage of DYRK1A, which is located on chromosome 21, is implicated in aberrant splicing associated with neurodegeneration in Downs syndrome [86]. Additionally, DYRK1A has been shown to play a role in homeostatic processes, such as lymphoid development and oncogenic pathways, such as DNA damage response, angiogenesis, and stem-like cell maintenance in neurological malignancy [87,88]. Pan-cancer analysis across The Cancer Genome Atlas (TCGA) indicates that many solid tumors and hematologic malignancies deregulate DYRK1A to promote tumor survival and growth. Research has shown that Cituvivint (SM08502), a pan-CLK/DYRK inhibitor, induced programmed cell death at concentrations which inhibited the accumulation of phosphorylated SR proteins in a panel of hematologic PDXs. Alternative splicing analysis showed that, of the drug-induced alternative splicing events, pathways known to drive hematopoietic lineages such as MAP kinase and mTOR signaling were enriched [89]. Interestingly, dual treatment with DYRK1A inhibitors and venetoclax synergize to increase cell death in several AML cell lines, while DYRK1A/CLK inhibition has also been proposed as a method to overcome venetoclax resistance in AML [90,91].

Another promising approach to targeting splicing involves targeting protein arginine methyltransferases (PRMTs), which catalyze arginine dimethylation on various substrates, including RBPs and splicing factors. PRMTs can be broadly categorized into Type-I (PRMT1, 2, 3, 4, 6, and 8) or Type-II PRMTs (PRMT5, 7, and 9) that catalyze asymmetric and symmetric arginine dimethylation, respectively. Among the major PRMT substrates are Sm proteins (B/B’, D1, and D3), which are important for spliceosome assembly and maturation. Deletion or chemical inhibition of PRMT5 causes hematopoietic failure and is linked to apoptosis, reduced quiescence, and inefficient splicing, resulting in intron retention and exon skipping in murine hematopoietic stem and progenitor cells [92]. The inhibition of PRMT5 has been shown to result in the aberrant splicing of genes associated with apoptosis and cell cycle in an Eμ-MYC-driven lymphoma model [93]. Additionally, pre-clinical studies showed that spliceosome-mutant leukemia cells show greater dependency on PRMT1 and PRMT5 [94]. Pre-clinical studies in *SF3B1* mutant uveal melanoma using PRT543, an oral PRMT5-selective inhibitor in conjunction with other therapeutic agents such as DNA alkylating agents or PARP inhibitors, resulted in synergistic reductions in cell viability [95]. This work formed the foundation of a phase I multicenter dose-escalation study to treat patients with *U2AF1* and *RBM10* mutant non-small-cell lung cancer (NCT03886831).

### 5.3. Targeting RNA Binding Proteins

Recently, the discovery that indisulam (E7070), an aryl sulfonamide-based anticancer compound, exerts its anti-tumor activity through the degradation of RBM39 [96,97] presents a new therapeutic opportunity to target splicing in cancer. Initial studies using indisulam reported that treatment in two colorectal carcinoma lines lead to G2/M-phase accumulation [98,99]. A subsequent analysis of indisulam’s effectiveness across 42 tumor types found that indisulam had a unique anti-proliferative spectrum with a wide range of IC50 values [100]. Since its development, indisulam has been tested in 11 clinical trials, both alone and in combination with other cancer therapeutics in solid tumors and hematologic malignancies to varying degrees of success.

Mechanistically, indisulam was found to induce RBM39 protein degradation through the proteosome by recruiting the CUL4-DCAF15 E3 ligase [96,97]. RBM39, also known as CAPER-α or HCC1, was first discovered in a chronic liver disease patient who later progressed to hepatocellular carcinoma [101]. RBM39 expression is tissue-specific—hematopoietic cells of myeloid and lymphoid lineages have the highest expression levels [102]. Increased RBM39 expression has also been identified across a host of solid tumors and hematologic malignancies [103]. A pan-cancer analysis of RBM39 correlates expression with clinical outcome for patients in terms of overall survival, dependent on tumor type [104]. Additionally, RBM39 has been proposed as a unique vulnerability in leukemias bearing splicing factor mutations [105].

RBM39 is an SR-related protein containing an N-terminus RS domain, two central RNA recognition motifs and a C-terminal U2AF homology motif (UHM), a specialized RNA recognition motif, which bind U2AF ligand motifs (ULMs). RBM39 has been shown to play a role in regulating transcription and alterative splicing [106,107]. Structural, localization, and pull-down assays have demonstrated that RBM39 interacts with U2AF1, U2AF2, and SF3B1, which is mediated by interactions between their UHM and ULM, respectively [106,107,108]. The splicing analysis of cells with RBM39 knockdown or treated with indisulam show that both inhibition or loss of RBM39 leads to a dramatic increase in exon skipping and intron retention [105].

### 5.4. Targeting Splicing Using Oligonucleotide-Based Therapy

As most broad-spectrum splicing inhibitors possess inherent risks relating to toxicity and off-target effects, there has been significant progress in developing alternative approaches to target pathogenic RNA splicing defects in diseases. Oligonucleotide-based modalities such as decoy oligonucleotides and antisense oligonucleotides (ASOs) offer the potential to target disease-causing isoforms with a high level of specificity. ASOs come in a variety of flavors based on chemical modifications, which can affect function (reviewed in [109]). Additionally, ASOs can be designed to pair with different regions of a transcript to selectively fine-tune splicing or trigger quality surveillance pathways to promote transcript degradation.

Using ASOs to modify splicing was first described in 1993 by Dominski and Kole—ASOs were used to correct pathogenic splicing events in vitro, which resulted from SNPs seen in β-thalassemia patients [110]. Mechanistically, splice-switch ASOs (SSOs) work by binding, via Watson–Crick base pairing to RNA sequences recognized by splicing machinery. SSOs can be designed to bind cryptic branch points, forcing the splicing machinery to use canonical 3’ splice sites, as in the case of β-thalassemia, or they can alter the inclusion or exclusion of exons by binding intronic or exonic splice enhancers or silencers. Since Dominski and Kole’s seminal work, SSOs have been tested in a variety of diseases to correct or alter splicing outcomes. We will highlight several recent applications as they pertain to the use of SSOs as cancer therapeutics. A more comprehensive summary on the use SSOs as splicing modulators across multiple human diseases is reviewed in [111].

A recent study published by Ma et al. used SSOs to correct pyruvate kinase (PK) isoform switching in hepatocellular carcinoma (HCC) [112]. In HCC, the pro-tumorigenic M2 isoform of PK (PKM2) is upregulated over the M1 isoform (PKM1) and promotes glucose uptake and lactate production in the presence of oxygen, a process known as aerobic glycolysis (known as the “Warburg effect”). These isoforms arise due to differential exon inclusion, where PKM1 includes exon 9 but not 10 and PKM2 includes exon 10 but not 9. In healthy cells, PK isoforms mark the stages of differentiation—PKM1 is expressed in more terminally differentiated cells and PKM2 is highly expressed in proliferating embryonic cells. Ma et al. reduced exon 10 inclusion by blocking the recognition of an ESE in exon 10 [112]. In vitro, both transfection and passive SSO uptake in several HCC cell lines led to a modest but significant increase in PKM1 levels and concomitant decrease in cell proliferation and increase in apoptosis. These results were recapitulated in vivo where systemic treatment or ectopic expression of the SSO reduced tumor volume and increased overall survival in PDX and murine HCC tumor models. This is merely one of many studies using SSOs to overcome pathogenic isoform changes arising from oncogenic transformation [113,114,115,116,117]. Moreover, SSOs have been used to target splicing changes resulting from drug resistance [57,118,119], and alter splicing to affect protein localization or isoform expression as a means of inhibiting tumor growth [120,121].

SSOs can also be used to reduce transcript levels—altering canonical splicing can shift a transcript’s reading frame to create PTC-containing species, which are then degraded by NMD. In a proof-of-concept study, Li et al. sought to reduce the level of ERG, an oncogenic driver in prostate cancer, using SSOs targeting exon 4 of ERG. In half of prostate cancers, ERG overexpression results from a 3Mb deletion, which fuses an androgen responsive promoter of *TMPRSS2* to exon 4 of *ERG* [122]. To target both the endogenous and fusion transcript for degradation, SSOs were designed to hybridize with 5’ or 3’ splice sites to block the binding of the U1 snRNP or U2AF heterodimer, respectively, leading to its exclusion. A loss of ERG exon 4 was predicted to create a PTC resulting from a shift in the reading frame, leading to ERG transcript degradation by NMD. Treatment with either 5’ or 3’ SSOs led to a modest but significant reduction in ERG4 mRNA and protein levels as well as reduced cellular proliferation and increased apoptosis in vitro. These results were recapitulated in vivo, where mice treated with the 3’ SSO showed a mild but significant reduction in tumor volume. Additionally, 3’ SSO-treated human prostatectomy cores had reduced ERG protein [115].

Correcting aberrant splicing resulting from splicing factor gene mutations is another potential avenue for the use of SSOs in cancer treatment. Inoue et al. investigated the consequences of BRD9 aberrant splicing, which reduced BRD9 expression via NMD, in *SF3B1*-mutant in uveal melanoma [19]. *SF3B1* mutation causes the use of a cryptic 3’ splice site, leading to the inclusion of exon 14a, termed a “poison exon” due to the fact that it contains a PTC. The inclusion of exon 14a leads to the degradation of the BRD9 transcript, resulting in increased cell proliferation and cytokine-independent growth of murine myeloid cells. Inoue et al. designed SSOs to block the 5’-splice site of exon 14a, preventing its inclusion, which restored BRD9 expression. SSO-treated SF3B1 mutant cells and PDX models had a significant reduction in proliferation and tumor volume.

Another method, which has yet to be tested but bears some consideration, is modulating pathogenic RNAs with ASOs, post-splicing. Since cancer alters splicing to produce RNA species not detected or detected as low levels in healthy cells, ASOs could be designed to leverage quality control mechanisms to target only aberrantly spliced transcripts. A paper published by Liang et al. demonstrated that ASOs can degrade cytoplasmic RNAs in vitro. In their proof-of-concept paper, the group tiled ASOs across the length of several mature transcripts and demonstrated that ASOs targeting 3’ regions of the RNA lead to an increase in lighter polysome fractions and with a concomitant reduction in target mRNA levels in a translation-dependent manner. Their work indicated that ASO-bound transcripts were targeted for degradation by no-go decay (NGD) [123]. NGD is a translation-dependent mRNA quality surveillance mechanism triggered by cytoplasmic RNAs with stalled and colliding ribosomes (reviewed in [124]). While this method for targeting RNA occurs independent of splicing, this could be a novel method to reduce pathogenic transcript levels resulting from alternative or aberrant splicing due to oncogenic transformation. While this method would require that pathogenic splicing occurs in 3’ regions of the transcript, it may present a novel method to specifically target cancer cells by leveraging oncogenic splicing aberrations.

In addition to targeting RNA, oligonucleotides can function as decoy molecules for splicing machinery. Denichenko et al. developed a novel method to perturb the splicing machinery by creating decoy nucleic acids designed to bind specific auxiliary splicing factors [125]. Upon hybridizing with their target proteins, these oligos inhibit protein-RNA binding while leaving RNA-independent cellular functions unperturbed. They selected four splicing factors, SRSF1, RBFOX1/2, and PTBP1, which are known to bind well-defined consensus mRNA sequences to modulate the alternative splicing of select transcripts. Sense oligonucleotides showed high specificity for their cognate protein and affected its ability to modulate splicing while leaving its ability to participate in RNA-independent cellular functions intact. Phenotypic analysis then linked splicing inhibition to published phenotypes for each protein. As an example, MKNK2, a SRSF1 target, activates the p38-MAPK stress response pathway, a tumor suppressive pathway. In vitro transfection of SRSF1 decoy oligonucleotides led to an increased phosphorylation of p38-MAPK targets and reduced oncogenic proliferation and anchorage-independent growth. In vivo, ectopic SRSF1 decoy oligonucleotide expression reduced murine tumor volume [119]. While this method is limited to splicing factors with highly conserved consensus motifs and the systemic administration of decoy oligos has yet to be tested, preliminary results from this paper indicate that this may be a promising method to target oncogenic splicing factors to modulate their function.

### 5.5. Immunotherapeutic Approaches Targeting RNA Splicing

The introduction of immune checkpoint blockade therapies has resulted in significant clinical improvement in various cancers. A major determinant of a positive response rate to immune checkpoint therapy is the amount of cancer-specific neoantigens, primarily driven by a higher tumor mutation load and acquired mutations in mismatched repair genes. In addition to neoantigens driven by somatic variants in coding genes, peptides derived from non-coding mutations and aberrant RNA splicing and processing can theoretically contribute to the pool of cancer-specific neoantigens. While multiple studies analyzing TCGA RNAseq data have concluded that tumor-specific alternative splicing is a regular occurrence and DNAseq data analysis has identified recurrent mutations in 119 splicing factors across both solid tumors and hematologic malignancies [126,127,128], few studies exist to determine if neoantigens resulting from tumor-derived splice events can be leveraged for the immunotherapeutic treatment of malignancy.

In a recent proof-of-concept study, Lu et al. sought to determine if splicing modulation could generate neoantigens, which elicited anti-tumor immunity [129]. This study was initiated when it was noticed that a sub-lethal treatment of cancer cells with indisulam resulted in sustained growth defects, following engraftment into mice, despite no substantial growth defects in vitro. Tumor volume could be further decreased by treating mice with anti-PD1 therapy, following the engraftment of tumors pretreated with indisulam. It was determined that this defect was T cell- and MHC I expression-dependent. In silico analysis of alternative splicing, proteomics, and MHC binding identified 109 candidate peptides resulting from indisulam treatment with predicted binding to one of the two murine MHC I haplotypes. The functional validation of these neoepitopes showed that 11 of these predicted events elicited a CD8+ T cell response in vitro and in vivo. Additionally, neoantigen-immunized T cells were able to recognize and selectively kill cancer cells pretreated with indisulam prior to transplantation. This study establishes that aberrantly spliced neo-peptides could potentially be exploited as strong inducers of the immune response. Currently, there is an emerging interest in developing improved neo-antigen identification technologies. Coupling this with rapid advances in cellular therapies such as chimeric antigen receptor (CAR) or synthetic T-cell receptor (TCR)-based therapies, bi-specific antibodies, and new generations of antibody–drug conjugates would lead to the development of promising immunotherapeutic modalities for precision anticancer therapies.

### 5.6. Synthetic Introns for Mutation-Specific Gene Expression

While small molecule-based splicing inhibitors and oligonucleotide-based approaches offer distinct advantages in targeting spliceosome-mutant cancers, their true effectiveness are significantly limited by the lack of target specificity and scalability. To overcome these challenges, a recent publication has suggested that leveraging the neomorphic effects of splicing mutations could be utilized as a mechanism to target cancer cells containing this mutation. North et al. hypothesized that mutant SF3B1-bearing cells could leverage this aberrant splicing activity to generate mutation-dependent protein products, which would allow for the selective killing of these cells by anti-viral medication [130] (Figure 5). The authors queried 20 cancer types with at least one SF3B1 mutation to identify intronic sequences, which were strongly predicted to be recurrently mis-spliced via the use of cryptic 3’ splice sites. Following validation using minigene luciferase splicing reactions, six candidate exons were inserted in the herpes simplex virus-thymidine kinase (HSV-TK). Treatment of HSV-TK expression cells with the antiviral prodrug ganciclovir causes cytotoxic metabolite production, leading to cell death. Further optimization through a series of mutations and deletions tested with massively parallel splice assays yielded an exogenous expression cassette, which preferentially killed SF3B1 mutant cells but not SF3B1 WT cells in the presence of ganciclovir both in vitro and in vivo.

## 6. Conclusions

Over the past decade, significant advances have been made in understanding how aberrant RNA splicing drives tumorigenesis. How cancer cells hijack the splicing machinery to propagate, survive, and withstand cancer therapies, however, requires further exploration. Novel developments in functional genomics (e.g., CRISPR/Cas-based technologies), chemogenomic, chemical biology, and synthetic biology tools will continue to guide the discovery of disease-related mechanisms and therapeutically actionable targets against aberrant RNA splicing in cancer.

## Figures and Tables

**Figure 1 genes-14-01378-f001:**
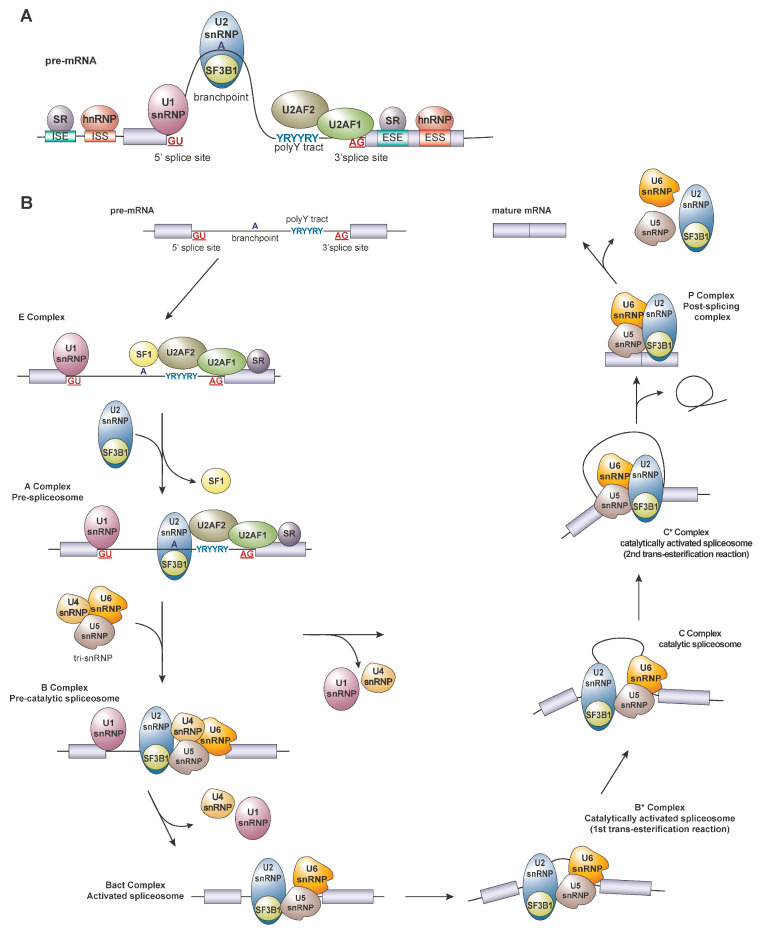
Spliceosome assembly and splicing catalysis. (**A**) Early spliceosome assembly: *Trans*-acting splicing factors and splicing regulators recognizing specific *cis*-elements on the pre-mRNA. (**B**) Formation of the E complex is formed by the recognition of the 5′ splice site (by U1 snRNP), branchpoint (by SF1), the polypyrimidine tract, and 3′ splice site (by the U2AF heterodimer). The U2 snRNP replaces SF1 and recognizes the branchpoint adenosine via base pairing, forming the A complex. The U4/5/6 tri-snRNP complex is recruited to the A complex to form the B complex (pre-activated spliceosome). Through a series of conformational changes that displaces the U1 and U4 snRNPs and results in the formation of the activated B (Bact) and catalytic B (B*) complexes, resulting in the first trans-esterification reaction step. This is followed by the formation of the activated C complex and the catalytic C complex (C*), which executes the second trans-esterification reaction. The completion of the cycle results in the formation of mature mRNAs via exon ligation and the release of the remaining splicing proteins and the intron lariat.

**Figure 2 genes-14-01378-f002:**
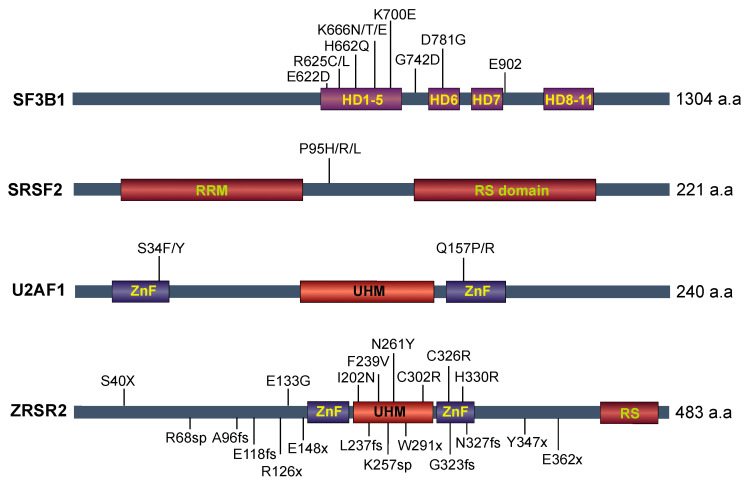
Somatic mutations in the four most commonly mutated spliceosome-associated proteins SF3B1, SRSF2, U2AF1, and ZRSR2. Plots showing the locations of recurrent mutations. HD—Heat domain; RRM—RNA recognition motif, RS—serine/arginine-rich; ZnF—zinc finger; UHM—U2AF homology motif; fs—frameshift; sp—splice site.

**Figure 3 genes-14-01378-f003:**
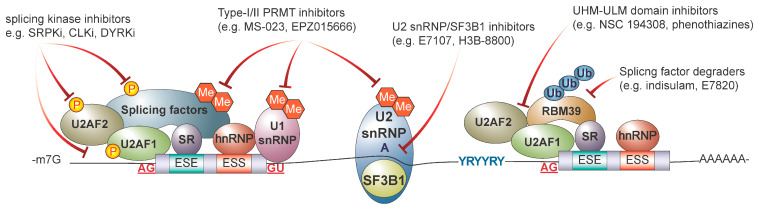
Small molecule approaches targeting RNA splicing. Diagrammatic depiction summarizing various therapeutic-targeting modalities against RNA splicing machinery. This includes targeting the splicing factors and splicing regulatory proteins using small molecules. P—phosphorylation; Me—methylation; Ub—ubiquitination.

**Figure 4 genes-14-01378-f004:**
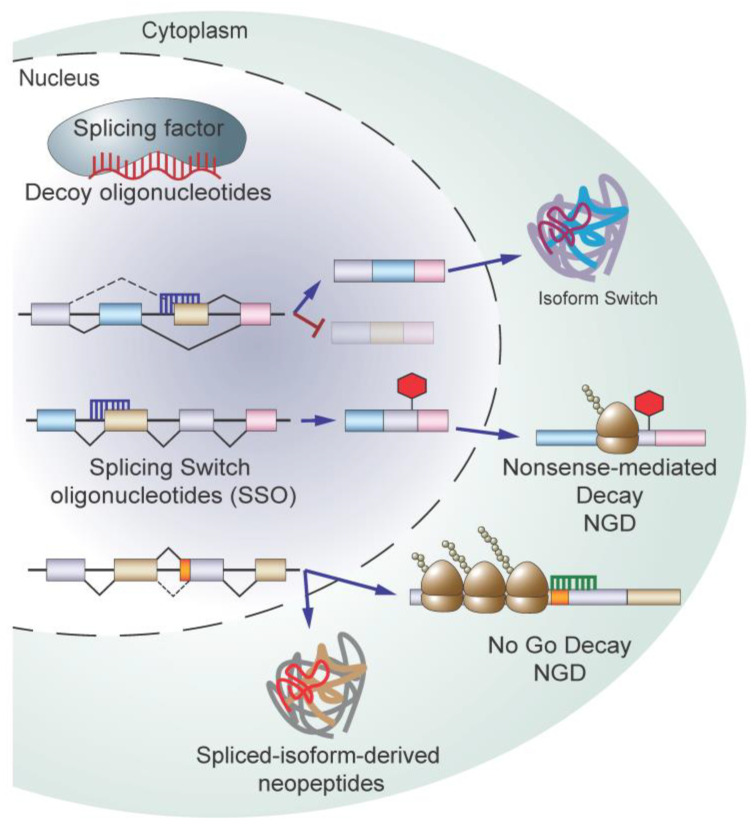
Oligonucleotide and immunological approaches targeting RNA splicing. Summary of various published methods of ASOs to target splicing machinery, including decoy ASOs, splice switch oligonucleotides (SSOs), and inducers of no-go mediated decay (NGD). Additionally, splicing-derived neoantigens are depicted (bottom).

**Figure 5 genes-14-01378-f005:**
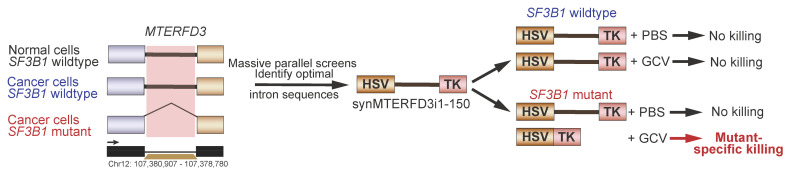
Synthetic introns as therapeutics in cancers with splicing factor mutations. A summary of methods used to develop and test synthetic introns. From left to right: MTERFD3 was identified as a strong candidate for the preliminary design of an optimal synthetic intron; massive parallel screening and mutagenesis were used to optimize the intronic sequence to generate synMTERFDi1-150; in vitro studies show that synthetic intron splicing occurs in *SF3B1* mutant cells, leading to HSVTK expression and cell death upon ganciclovir (GCV) treatment.

**Table 1 genes-14-01378-t001:** Splicing factor alterations in cancer.

Organ	Splicing Factor	Type of Alterations
Brain	*SRSF1, SRSF3, HNRNPA1, HNRNPA2, HNRNPHK*	Upregulation
Breast	*SRSF1, SRSF3, SRSF4, SRSF5, SRSF6, TRA2B, HNRNPA1, HNRNPI* *RBM5, RBFOX2, HNRNPK* *SF3B1*	Upregulation Downregulation Somatic mutation
Bladder	*SRSF1, SRSF3*	Upregulation
Colon	*SRSF1, SRSF3, SRSF6, SRSF10, TRA2B* *HNRNPK, RBFOX2*	Upregulation Downregulation
Intestine	*SRSF1*	Upregulation
Kidney	*SRSF1, SRSF3*	Upregulation
Liver	*SRSF3*	Upregulation
Lung	*SRSF1, SRSF3, SRSF5, SRSF6, TRA2B* *RBM5, QKI* *RBM10, U2AF1*	Upregulation Downregulation Somatic mutation
Skin	*SRSF3* *HNRNPK* *SF3B1, SRSF2*	Upregulation Downregulation Somatic mutation
Thyroid	*SRSF1, SRSF3* *RBM10*	Upregulation Somatic mutation
Myeloid leukemias	*SF3B1, SRSF2, U2AF1, ZRSR2*	Somatic mutation
Chronic lymphocytic leukemia	*SF3B1*	Somatic mutation

**Table 2 genes-14-01378-t002:** Alternative splicing events associated with resistance to cancer therapy.

Gene	Splice Variant	AS Event	Therapeutic Resistance	Cancer	Mechanism	Ref.
*AR*	AR-V7	Cryptic exon usage	Androgen deprivation therapy	Prostate	Removes ligand-binding domain	[49]
*BIM*			Tyrosine kinase inhibitors	Lung		
*BRACA1*	BRCA1-Δ11q	Alternative SS usage	PARP inhibitors	Breast, ovarian	Mutation removal	[50]
*BARD1*	BARD1β	Exon 2,3 skipping	PARP inhibitors	Colon	Prevents BARD1/BRACA1 dimerization	[51]
*BRAF*	p61BRAF(V600E)	Exon 4-8 skipping	BRAF inhibitors	Thyroid, Skin	Inhibits downstream signaling	[52,53]
*HER2*	ΔHER2	Exon 16 skipping	mABs	Breast	Homodimer stabilization	[54]
*MS4A1*	CD20-V1, -V2	Aberrant 5’ UTR splicing	mABs	Lymphoma, Leukemia	Translation inhibition	[55]
*TAK1*	TAK1ΔE12	Exon 12 skipping	Chemotherapy	Breast	JNK/p38 activation	[56]
*PKM*	PKM2	Isoform switch	Chemotherapy	Pancreatic	Unknown	[57]

## Data Availability

Data sharing is not applicable to this article.

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
