# Peer review of "Therapeutic Targeting of RNA Splicing in Cancer"

_genes, 2023, doi:10.3390/genes14071378_

Round 1

Reviewer 1 Report

This review manuscript focuses on anti-cancer therapeutics targeting splicing, from small molecules to antisense oligonucleotides to gene therapy-based approaches. This manuscript provides a useful update on some topics, such as the spliceosome targeting compounds which used to hold promise but failed in recent trials. The cited papers are ok, yet I think that there should be more citations of the original articles (not simply rely on other reviews), so as to give a broader visibility to other studies that cannot be mentioned in detail in this paper due to space constraints. These citations should help the reader locate the relevant information, which should be one of the main purposes of review papers like this one. Thus, this manuscript only needs minor corrections as detailed below:

1.       Chapters 3 and 4 outline the cancer-associated changes in expression of splicing factors and their mutations. While this is discussed in less detail than the therapeutics, some statements are not backed by citations. First, on page 4, BIN1, RON and BIM have interesting alternative splicing events whose original citations should be added. Even splicing events in UTRs might drive tumorigenesis, according to a recent paper. Second, section 4.5 should provide more information about other splicing factors such as LUC7L proteins, and original citations for the already mentioned splicing factors (RBM10, PRPF8, SRRM2, etc).

2.       In few cancers, splicing events and factors are not associated with tumorigenesis, but instead with drug resistance (like imatinib in CML) and metastasis (recent RBFOX2 paper in Nature), and this should be discussed too.

3.       For SSOs, there are more cases in cancer such as BIM, KDR, STAT3. Briefly mention and cite.

4.       On page 1 subheading 2, “Regulation of splicing catalysis” is misleading because regulation happens well before the two catalytic steps, delete the word ‘catalysis’.

5.       In the first sentence after this, exon ligation does not happen after (subsequently) intron removal, instead it happens at the same time. Right after, the spliceosome also has individual polypeptides in addition to snRNAs and snRNPs.

6.       On page 2, the spliceosome assembly pathway is missing complexes Bact and C*.

7.       Figure 2, these splicing factors have many more mutations than the ones shown.

8.       Section 5.2 is a bit sloppy, the SR protein kinases SRPKs, instead the letters are scrambled to SPRK.

9.       Page 9 line 326, do “nuclear patches” refer to “nuclear speckles” instead?

10.   Page 11 line 431, U2AF is a heterodimer, not a homodimer.

11.   Page 12 lines 471-474, only RBFOX1/2 bind to a non-degenerate RNA consensus sequence, the others do not. 

12.   Last, this manuscript will benefit from language editing to fix grammar. 

Reviewer 2 Report

General comments: 

-       Well-written, thorough, up-to-date review and insightful review on the therapeutic targeting of splicing defects in cancers. 

-       Authors thoroughly describe, in most cases, how specific splicing inhibitors prevent tumorigenesis. It sounds important to discuss how/why these treatments differentially influence the biology of cancerous and non-cancerous cells. It is touched upon in some cases, but I think it could be more addressed more specifically throughout the manuscript.

-       It would be worth mentioning that U2 snRNP-interfering activity of SSA also promotes premature cleavage and polyadenylation by limiting U1 snRNP availability and thereby inhibiting telescripting (Yoshimoto et al., 2021, Cell Chemical Biology). Authors should also review important indirect effects of other splicing inhibitors on gene expression since they may be the cause of 

-       There are no references to Figures 3, 4 and 5 in the text.

Minor comments:

Line 8: Splicing is largely co- and not post-transcriptional.

Line 45: The described choreography of events does not include the characterization of the B complex.

Lines 89-91: It would be interesting to clarify how the splicing factor SRSF1 functions with MYC to activate mTOR. For example, does it involve a role for MYC in splicing, a role for SRSF1 in transcription, or independent functions of SRSF1 and MYC in splicing and transcription that converge towards mTOR activation?

Lines 106-107: Something is missing in this sentence.

Line 114: Please describe the acronym used for each domain in the legend. For example, does HD stand for “Heat repeat domain”? It’s unclear since those domains are called “HRs” later in the text. Additionally, what are the labeled resides mutated to? Are the resulting mutated amino acids different in different patients?

Lines 119-120: Given what is written in section 2 (lines 45-54), it would be helpful to clarify that SF3B1 recognizes the 3’-splice site during B complex and not E complex formation. Additionally, what mutations are the authors referring to? Any of those shown in Figure 2? Probably not since the next paragraph distinguishes different groups of mutations.

Line 142: Is missing before “metabolism,”?

Line 155: Does Q157 promote or repress exon inclusion when 3-SSs are followed by G over A?

Line 173: Are ZRSR2 mutations indels? How else would it shift the reading frame? Were all the above-discussed mutations only base substitutions? It could be clarified throughout the manuscript.

Line 179: U snRNA genes are intronless, so what do they author mean by “U12 intron retention” and “U2 intron splicing”?

Line 198: Maybe authors should revisit this title since it includes inhibitors that do not target SF3B.

Line 235: What is PDX?

Lines 237-238: How do “U2 snRNP components” differ from “U2 snRNP-specific proteins”?

Lines 232-241: It would worth emphasizing that pladienolides B is commonly known as PladB and widely used, like SSA, in splicing studies. Like for the other SF3B-targeting drugs, it would be interesting to discuss how PladB limits tumorigenesis.

Lines 382-384: UHMs/ULMs were described earlier in the manuscript (lines 276-278). Lines 276-278 should also be where authors specify how ULMs are “non-RNA binding” “RNA-binding motifs”.

Lines 411-413: Something is wrong with this sentence.

Line 426: There are numerous RNA quality surveillance pathways, authors should thus specify that they are referring to NMD.

Lines 427-428: I think it would help to specify that ERG4 is a splicing isoform of the ERG gene. I have worked on this gene, and it took me quite some time to figure out what this paragraph was about.

Line 444: Do the authors mean degradation in the cytoplasm by NMD?

Line 456: Do the authors mean a relocalization of the targeted mRNA from heavy to light polysome fractions, or a global increase of light polysomes in cells?

Lines 515-518: The concept that inducing aberrant splicing can be in tumor immunotherapy is very interesting. Thus, it would be beneficial to clarify what the other mentioned cellular therapies are and how their coupling to neo-antigen presentation technologies could bring this field further.

Reviewer 3 Report

The review paper clearly described current progress of RNA splicing mechanisms, RNA splicing in cancer, and recent therapeutic targeting of RNA splicing. The paper covers all of the recent progress in the field. Writing is clear, figures are well drawn.

The authors summarized recent progress in splicing regulatory mechanisms, mis-splicing in tumor initiation and progression including altered splicing factor expression in cancer, recurrent mutations in splicing factors in cancer. Importantly, they addressed therapeutic targeting of RNA splicing in cancer, especially core spliceosome with SF3B compounds, splicing regulatory proteins, RNA binding proteins, ASO-based therapy, immuno-therapeutic approaches targeting RNA splicing, synthetic introns for mutation-specific gene expression. The manuscript is well written.

Specific comments:

Crisp/cas9 system is a promising therapeutic tool. If the tool has already been used in cancer therapy, the authors can cite the references. If not yet, the authors can briefly describe in the conclusion part.
